# *Salmonella enterica* Serovar *Dublin* from Cattle in California from 1993–2019: Antimicrobial Resistance Trends of Clinical Relevance

**DOI:** 10.3390/antibiotics11081110

**Published:** 2022-08-17

**Authors:** Heather M. Fritz, Richard V. Pereira, Kathy Toohey-Kurth, Edie Marshall, Jenna Tucker, Kristin A. Clothier

**Affiliations:** 1California Animal Health and Food Safety Laboratory, School of Veterinary Medicine, University of California, Davis, CA 95616, USA; 2Department of Population Health and Reproduction, School of Veterinary Medicine, University of California, Davis, CA 95616, USA; 3California Animal Health and Food Safety Laboratory, School of Veterinary Medicine, University of California, San Bernadino, CA 92411, USA; 4California Department of Food and Agriculture, Antimicrobial Use and Stewardship, Sacramento, CA 95814, USA

**Keywords:** *Salmonella enterica* serovar *Dublin*, cattle, antimicrobial resistance, trends, antimicrobial susceptibility testing, MIC, NARMS

## Abstract

*Salmonella enterica* subsp. *enterica* serovar *Dublin* (*S.*
*Dublin*) is a cattle-adapted pathogen that has emerged as one of the most commonly isolated and multidrug resistant (MDR) serovars in cattle. *S.*
*Dublin* may be shed in feces, milk, and colostrum and persist in asymptomatic cattle, leading to spread and outbreaks in herds. Though infections with *S.*
*Dublin* in humans are rare, they are frequently severe, with extraintestinal spread that requires hospitalization and antimicrobial therapy. To determine minimum inhibitory concentration (MIC) and antimicrobial resistance (AMR) patterns and trends in cattle in California, broth microdilution testing was performed on 247 clinical *S*. *Dublin* isolates recovered from cattle at the California Animal Health and Food Safety Laboratory System (CAHFS) over the last three decades (1993–2019). Mean MICs and classification of resistance to antimicrobial drugs using a clinical livestock panel and the National Antimicrobial Resistance Monitoring System (NARMS) Gram-negative drug panels were utilized to assess prevalence and trends in AMR. Findings indicate an increase in AMR for the years 1993 to 2015. Notably, compared to the baseline year interval (1993–1999), there was an increase in resistance among quinolone and cephalosporin drugs, as well as an increased number of isolates with an MDR profile.

## 1. Introduction

*Salmonella enterica* subsp. *enterica* serovar *Dublin* (*S.*
*Dublin*) is a cattle-adapted serovar that causes significant morbidity and mortality in juvenile and adult animals [1,2]. Though infrequent, infection in humans can occur and often leads to clinically severe infection, with septicemia and death as common sequelae [3]. In recent years *S.*
*Dublin* has emerged as one of the most frequently isolated serotypes of *Salmonella* in cattle [2]. Infected animals may shed organisms in feces, milk and colostrum, and chronic asymptomatic shedding may occur in carrier animals [4]. Once in the environment, *S*. *Dublin* can persist for months to years [1,5,6]. There is an increasing prevalence of *S.*
*Dublin* reported in both cattle and humans, as well as multidrug resistance (MDR) detected in these isolates [3,7]. As a host-adapted serovar in cattle, *S.*
*Dublin* is highly invasive and commonly associated with septicemia [8]. Multiple reports suggest that the prevalence of infection, the risk for asymptomatic carrier state with organism shedding and the higher levels of antimicrobial resistance (AMR) are greater in young dairy cattle as compared to adult cattle and beef cattle operations [4,7,9]. Antibiotic treatment is controversial, as *Salmonella* spp. are frequently resistant to many antibiotics approved for use and antibiotic exposure may lead to the development of drug resistance in non-target organisms. However, systemic infections and infections in young calves carry a high risk of mortality without the timely administration of effective antibiotics [2].

The increasing threat that antimicrobial resistance poses to both human and animal health has led to an abundance of multilevel national and international efforts to evaluate, monitor and curb antimicrobial resistance. The 2015 National Antimicrobial Resistance Monitoring System (NARMS) Integrated Report indicates increasing MDR in *S.*
*Dublin* with a high level of resistance to ceftriaxone and decreased susceptibility to ciprofloxacin, which are critically important for treating severe *Salmonella* infections in humans [10]. Drug-resistant non-typhoidal (NT) *Salmonella* was listed as one of the Serious Threats in the 2019 National Strategy for Combating Antibiotic-Resistant Bacteria (CARB) Antibiotic Threats in the United States report [11]. Along with *Campylobacter*, NT-*Salmonella enterica* are the most frequent bacteria causing foodborne illness in humans in the United States, with an estimated 1.35 million illnesses and 420 deaths caused each year by *Salmonella* [12].

The extensive use of antibiotics in food-producing animals is often implicated as one of the major driving forces for the global trends in antimicrobial resistance [13]. To address this concern, recent legislative actions have led to the increased oversight and regulation of antimicrobials used in food-producing animals, including the categorization and restricted use of medically important antimicrobial drugs (MIADs) by the United States Food and Drug Administration (FDA). In 2017, the FDA implemented the Veterinary Feed Directive (VFD) to require oversight by a licensed veterinarian and a VFD order for all MIADs used in feed or a prescription for MIAD use in water [14]. California legislation (Livestock: Use of Antimicrobial Drugs, Food and Agricultural Code, FAC 14,400–14,408) enacted in January, 2018, expanded upon these requirements to ensure that any form of administration of a MIAD to livestock is performed in the context of a veterinarian–client–patient relationship [15]. 

Information on the long-term patterns of AMR in pathogens of critical importance, such as *Salmonella*, is scarce, particularly among livestock species. The purpose of this study was to evaluate the minimum inhibitory concentrations (MICs) and antimicrobial susceptibility trends to antimicrobials of clinical relevance to both veterinary and human medicine in *S.*
*Dublin* isolates recovered from cattle in California over a 27-year timeframe. 

## 2. Results

### 2.1. Sample Selection and Metadata Collected

A total of 247 *S.*
*Dublin* isolates collected from clinical samples received at the California Animal Health and Food Safety Laboratory System (CAHFS) between the years 1993 and 2019 were included in this study. We aimed to test approximately 10 *S. *Dublin** isolates per study period year from geographically diverse locations across the state of California. Therefore, isolates were selected so that for each year, individual isolates originated from a unique county and/or farm. Isolates were classified into the following response levels: year group (1993–1999; 2000–2005; 2006–2010; 2011–2015; 2016–2019), age group (early preweaned (PW); late PW; early heifer (HF); late HF; adult cow), geographic region (north, central, south), clinical signs (diarrhea vs. systemic illness) and season (winter, spring, summer, fall). The number of isolates in each response level category is shown in Table 1. 

The six different categories used for the “age group” variable refer to the age of the animal when *Salmonella* was isolated and are as follows: early PW, ≤4 weeks of age; late PW, >4 and ≤9 weeks of age; early HF, >9 weeks and ≤12 months of age; late HF, >12 and ≤17 months of age; adult cow, >17 months of age. The selected age cut-offs were based on common dairy cattle management practices that vary by age group, as well as references used in the National Animal Health Monitoring System (NAHMS) [16]. The “source group” classification was assigned according to the tissue from which *Salmonella* was isolated; for source groups where only two or fewer total isolates were obtained, (kidney, spleen, brain, colostrum, milk, ear and joint fluid), samples were clustered into the category “other”. The list of counties categorized into each region of California is provided in Appendix A. Clinical signs were either categorized as a systemic illness (when the isolate was recovered from internal organ site at necropsy and was associated with disease) or diarrhea (when only feces were submitted for testing and diarrhea was listed as a symptom on the submission form). Clinical history and/or final diagnosis were not available for 23 of the isolates tested and those were categorized as “unknown” for clinical signs. 

For all isolates, antimicrobial susceptibility testing (AST) was performed on both the commercial clinical BOPO6F livestock panel of drugs (Sensititre™ Bovine/Porcine BOPO6F Vet AST Plate, Thermo Scientific, Waltham, MA, USA) and the NARMS Gram-negative panel of drugs (Sensititre ™ NARMS Gram-Negative CMV4AGNF AST Plate). 

### 2.2. Minimum Inhibitory Concentration Distributions

The distributions of MICs for all isolates tested on the BOPO6F and the NARMS panels are shown in Table 2 and Table 3, respectively. For drugs with available clinical breakpoints, the breakpoints and proportion of isolates classified as resistant are also shown. No clinical breakpoints are available for *S.*
*Dublin* in cattle, but interpretations extrapolated from *Escherichia coli* in cattle from metritis and mastitis infections are available for ampicillin and ceftiofur, respectively, and were extrapolated as breakpoints for *S.*
*Dublin* in this study. Using these criteria, 97.1% of all isolates were classified as resistant to ampicillin and 46.1% were resistant to ceftiofur.

NARMS interpretations are intended for AMR monitoring and surveillance, and are not necessarily clinically relevant to cattle, but the panel was included in this study because it enabled the use of interpretations for analysis and generated data that are comparable to what is considered one of the best standardized approaches for monitoring AMR, as used by the Centers for Disease Control (CDC) NARMS for surveillance of enteric bacteria [17,18]. Compared to the BOPO6F panel, when using the NARMS drug concentrations and breakpoints, only 67% of isolates were classified as resistant to Ampicillin. The discrepancy in the proportion of isolates classified as resistant is a result of the different breakpoints used for human and NARMS surveillance panels when compared to veterinary-specific breakpoints for resistance. Using NARMS criteria, the drugs with the highest levels of resistance across all isolates tested were streptomycin, tetracycline, chloramphenicol, ampicillin, and sulfisoxazole (67–81% resistant). Additionally, at least 45% of all isolates tested were resistant to nine of the drugs on that panel (Table 3). 

### 2.3. Epidemiological Cut-Offs for MICs

For drugs that met the criteria for setting epidemiological cut-offs (ECOFFs), results are shown in Table 4. ECOFFs utilize MIC distribution data to separate bacterial populations into wild type (WT) and non-wild type, which are those that exhibit phenotypic evidence for acquired resistance [19]. On the BOPO6F panel, only tulathromycin fulfilled the criteria requirements for ECOFF designation. The ECOFF for tulathromycin fell at the highest dilutions tested on the panel, indicating that in this population of *S.*
*Dublin* tested, acquired resistance to this drug was not evident. Six drugs on the NARMS panel met the criteria for ECOFF designation and all but one, ceftriaxone, were 2–4 two-fold dilutions lower than the NARMS breakpoint for resistance. The 95% and 99% endpoint ECOFFs for ceftriaxone both fell in the NARMS-classified region of resistant concentrations, indicating agreement between the two methods for predicting resistance.

### 2.4. Proportion of Resistant Isolates over Time Using NARMS Criteria

Using resistance criteria from the NARMS panel, the proportions of isolates resistant to selected drugs was compared across year groups using the least square means and non-parametric Wilcoxon test pairwise analysis to identify significant differences. For the cephalosporin drugs (cefoxitin, ceftiofur and ceftriaxone), the proportion of *S.*
*Dublin* isolates that were resistant had a statistically significant increase with each year interval until the highest proportion resistant was reached for the year interval 2006–2010, after which the proportion resistant did not significantly change (Figure 1a).

The proportion of isolates resistant to the quinolones nalidixic acid and ciprofloxacin demonstrated a significant increase in the year interval 2011–2015 compared to 2000–2005 and again in the 2016–2019 interval compared to the 2006–2010 and 2000–2005 intervals (Figure 1b).

For the aminoglycoside gentamicin, the proportion of resistant isolates increased (though not significantly between year intervals) from 1993–2010, with highest proportion classified as resistant at 54% in 2000–2005. A significant decrease in resistance was observed in the year group 2011–2015 (15% resistant) compared to prior year intervals, and the proportion of isolates classified as resistant further decreased to less than 4% in the 2016–2019 year interval (Figure 1c).

### 2.5. Odds Ratios for Effect of Response Levels on Resistance Using NARMS Criteria

Resistance criteria from the NARMS panel was used in the final model output for the odds ratio (OR) comparison of resistance by response level for each drug, as well as for multidrug resistance (MDR). Here MDR was defined as resistance to three or more classes of antimicrobial drugs. The OR output is shown in Table 5, Table 6, Table 7, Table 8 and Table 9.

For all three of the cephalosporins (ceftiofur, ceftriaxone and cefoxitin), there were significantly lower odds of resistance in early years compared to later years until 2006–2010, after which year group comparisons were no longer significant (Table 5). Increased risk for resistance to tetracyclines was seen for the 2006–2010 and 2011–2015 year intervals when compared to 1993–1999 isolates; other year group comparisons for resistance to tetracycline did not significantly differ (Table 6). For gentamicin, there was a higher risk for resistance in 2000–2005 compared to 1993–1999, but the odds for resistance in the 2016–2019 time interval was significantly lower than 1993–1999 (Table 6). Notably, the odds for gentamicin resistance were significantly higher in the 2000–2005 year group compared to the later year groups 2011–2015 and 2016–2019 (Table 6). The odds ratio for resistance for nalidixic acid showed a significant shift towards increasing odds for resistance in later year intervals beginning in 2011–2015 (Table 7). Amoxicillin–clavulanic acid showed decreased odds for resistance in the year intervals 1993–1999 and 2000–2005 compared to later time points; but, as of 2006–2010, the odds ratio did not differ significantly with later year intervals (Table 7). There were decreased odds for resistance to chloramphenicol in the 1993–1999 interval compared to all later time intervals and also between 2000–2005 vs. 2011–2015 (Table 8). 

There was a significantly lower OR for MDR in the early years 1993–1999 compared to all later year intervals (Table 9). When season is taken into account, the only drug that differed significantly in OR for resistance was chloramphenicol, where the OR for resistance was higher in the fall compared to spring and summer (Table 8).

Regional comparisons demonstrated that, with the exception of gentamicin, resistance to all drugs retained in the analysis and the odds for MDR was higher in the central region compared to north. The odds for resistance in the south was significantly higher compared to the north. South and central California regions did not differ significantly in OR for resistance. For gentamicin, the OR for resistance did not differ significantly between central and north regions, but the OR for resistance was higher in the south compared to both north and central regions. 

Based on age group, early PW calves and early HF had a significantly higher risk for MDR than adult cows. The only drug that showed a significantly increased risk for resistance in early PW calves and early heifers, as compared to cows, was tetracycline, which was retained by the multiple Pearson chi-square test (Table 6).

## 3. Discussion

Because clinical breakpoints are not available for *S.*
*Dublin* in cattle, NARMS interpretive criteria were used to interrogate the trends and patterns of AMR in the study samples. While NARMS interpretations may not accurately reflect clinical efficacy of the drug in the bovine host, monitoring trends and risk factors for resistance and MDR using these available criteria provides information inherent to the bacteria that reveals a trend in decreasing antimicrobial susceptibility that likely is matched by decreasing clinical drug efficacy. The results of this study reveal a trend toward increasing AMR for *S.*
*Dublin* for the majority of drugs evaluated, as well as increased risk for MDR in later years compared to the baseline 1993–1999 year interval. A number of factors may contribute to the observed changes in resistance; those considered most significant are discussed below. 

Depending on the interpretation criteria used, 67% or 97% of *S.*
*Dublin* isolates were classified as resistant to ampicillin. A significant increase in OR for resistance was found for the year interval 2006–2010 compared to the baseline 1993–1999 interval, but no other year group comparisons yielded significant differences. Similar to ampicillin, the majority of isolates (76%) were resistant to tetracycline, with increased odds for resistance seen for 2006–2015 when compared to the baseline 1993–1999 interval. As will be discussed below for MDR in *Salmonella*, ampicillin and tetracycline resistance is commonly reported among *S.*
*Dublin*, and trends in mean MIC will be presented in the accompanying second manuscript. For amoxicillin/clavulanic acid (AMC), the odds for resistance were higher in all later year intervals, compared both to the baseline 1993–1999 year group and also to all study years preceding 2011–2019, at which point the odds for resistance did not differ from 2006–2010 year interval. 

For year interval comparisons, significant increases in odds for resistance to cephalosporins were found over time from the 1993–1999 year interval until 2006–2010, after which comparisons were no longer significantly different. This trend is similar to what was found for AMC. According to the 2015 NARMS Human Isolates Surveillance report, 67% of *Dublin* isolates were resistant to ceftriaxone [20], similar to the proportion of resistant isolates in this sample set for the third-generation cephalosporins tested (ceftiofur and ceftriaxone) for 2006–2019. The 2012 order issued by the FDA’s Center for Veterinary Medicine to prohibit the extra label use of cephalosporin drugs in cattle (FDA Extra label Use and Antimicrobials) [21] may explain why levels of resistance did not continue to rise in later year intervals in this study. It is worth noting, however, that while a continued increase in resistance was not evident, the proportion of isolates resistant to cephalosporins also did not significantly decrease in more recent study year intervals following the 2012 regulatory change in the use of these drugs. 

Chloramphenicol use is prohibited in food-producing animals, but its fluorinated derivative, florfenicol, is widely used to treat a variety of diseases in cattle and is approved to treat bovine respiratory disease. A number of genes have been shown to mediate resistance in both florfenicol and chloramphenicol [22], and the same genes may confer resistance for both [23]. It has also been shown that some bacterial isolates that harbor resistance genes, and are phenotypically resistant to chloramphenicol, are susceptible to the fluorinated derivatives. Additionally, the mechanisms for co-selection of resistance genes may occur in the absence of selection pressure imposed by use of chloramphenicol or florfenicol [24]. In the absence of clinical breakpoints to florfenicol in cattle, it is impossible to determine if the chloramphenicol resistance approximates florfenicol resistance. In the second accompanying manuscript, trends in MIC values are applied to glean information about susceptibility for drugs on the bovine clinical BOPO6F panel. 

The odds ratio for resistance to chloramphenicol was higher for the fall season compared to winter or summer. It is possible that this finding coincides with a history of increased prevalence and treatment of bovine respiratory disease (BRD), for which florfenicol is widely used. A recent study examining the epidemiology of BRD in preweaned calves in CA dairies found a higher prevalence for BRD in the fall [25].

Significant increases in resistance to nalidixic acid and ciprofloxacin were detected in isolates from 2011–2019. Nalidixic acid is a synthetic quinolone used for surveillance purposes to detect diminished fluoroquinolone susceptibility [26]. Interestingly, shifts in increasing MIC for nalidixic acid preceded those seen in the fluoroquinolones tested on both the BOPO6F and NARMS panels. The value of monitoring changes in MIC (that are not necessarily reflected by detected changes in measured resistance) will be discussed in the second accompanying paper. 

Because *S*. *Dublin* infections frequently result in systemic disease that commonly includes respiratory signs, it is possible that empirical treatment is being misdirected at bacterial organisms commonly associated with bovine respiratory disease (BRD), namely *Pasteurella multocida*, *Mannhemia haemolytica*, and *Histophilus somni*. Florfenicol, fluoroquinolones, macrolides, and cephalosporins (ceftiofur) might be selected, as labelled treatment options, and contribute to increased resistance in *S.*
*Dublin*. 

The rapid reversal of resistance seen with gentamicin, apparently preceding downward trends in other antimicrobials and greater than the changes observed for other aminoglycoside drugs in the panels, suggests an abrupt and widespread decrease in its use in California. It is possible that the extra label use of parenteral gentamicin in cattle had previously been more widespread but essentially came to a halt as FDA, the American Veterinary Medical Association, and others issued warnings about misuse and residues in the late 1990s and early millennium. [27,28]. Around that time there were FDA warnings against its illegal use [27]. Furthermore, there were emerging data from the Food Animal Residue Avoidance Databank (FARAD), showing the prolonged detection of aminoglycoside residues that posed a risk for violative residues in food animal tissues and the prohibition of extra label use of drugs in feed by the Animal Medicinal Drug Use Clarification Act (AMDUCA) [1]. Interestingly, the OR for resistance for gentamicin was higher in the south compared to both north and central regions, possibly reflective of regional practices, perceptions, or adherence to new restrictions; or animal products from those with higher resistance to gentamicin may have been family animals or otherwise not destined for human consumption.

Increased levels of MDR were observed for all time intervals when compared to the baseline year interval 1993–1999. This is similar to what was reported for *S*. *Dublin* for the year interval 1996–2004 compared to 2005–2013 [3] and comparisons of 2002–2009 and 2010–2016 *S*. *Dublin* isolates from dairy cattle in California that similarly showed an increase in MDR [7]. The whole genome sequencing of *S.*
*Dublin* isolates points to a plasmid-mediated acquisition of MDR [29,30,31]. Plasmid-mediated MDR has been described elsewhere as well [24]. Significant differences in phenotypic AMR profiles and the presence/absence of plasmid replicons were found by geographic location [31], which aligns with the findings of a study in Denmark that utilized WGS data to describe the epidemiology of *S*. *Dublin* revealing persistence of a single strain within herds for years [30]. The WGS of the isolates in this study, which is currently in progress, would further elucidate the basis of phenotypic resistance and any patterns related to geography or the year that the isolate was recovered. 

Frequently encountered MDR patterns reported for *S.*
*Dublin* isolated from cattle include ampicillin, chloramphenicol, streptomycin, sulfonamides, and tetracycline, with increasing levels of ceftriaxone resistance in more recent years [20,32]. Utilizing NARMS criteria, 67% of all isolates were resistant to sulfisoxazole but only 3.2% were resistant to TMS (Table 3 For the other drugs typically associated with MDR in *S. Dublin* listed above, between 67% and 81% of isolates in this study were classified as resistant.

In November 2019, an outbreak of *S.*
*Dublin* in humans occurred that was traced back to contaminated ground beef from California. This recent outbreak highlights the seriousness of risk to humans: 82% of sick individuals required hospitalization and one person died; and *S.*
*Dublin* was recovered from the blood of 46% of infected individuals [33]. In addition, the outbreak led to the recall of more than 34,000 pounds of ground beef, also highlighting the economic and food safety impacts from this serovar. Interestingly, whole-genome sequencing on the outbreak isolates did not identify any known antibiotic resistance mechanisms. AMR has been reported to be higher in clinical vs. non-clinical infections in cattle [26], which might explain this finding. A study that compared AMR between bovine and human isolates of *S.*
*Dublin* found that resistance was generally higher in bovine isolates and was associated with the presence of an IncA/C2 plasmid that genetically distinguished the bovine from human isolates [34].

With the exception of gentamicin, the odds ratio for resistance to each of the drugs tested on the NARMS panel was higher for central and southern California regions compared to northern California. Possible explanations for this include differences in the sizes and numbers of production facilities distributed throughout California, with larger numbers of large-scale productions being located in central and southern California [35]. It is also possible that regional variation in production types and their respective management practices regionally influence antimicrobial exposure and selection pressures for resistance. As an example, counties within the San Joaquin Valley, (the majority of which are in the central and southern regions of California), support the largest dairy productions in the state [35].

Increased odds for resistance to tetracycline, streptomycin, and MDR were encountered for isolates obtained from animals in the Early PW and Early HF age groups compared to adult cows. Because younger animals are more susceptible to a variety of neonatal disease agents, the increased therapeutic use of antimicrobials in this age group may account for some of the higher levels of resistance seen in younger animals. As reviewed by Springer et al., increased risk for harboring resistant *Salmonella enterica* and *Escherichia coli* among younger calves has been documented in numerous studies; however, the factors that contribute to the relative resistance remain poorly understood [36].

The present study has several limitations. *S.*
*Dublin* isolates originated from sick or dead animals, creating a biased sampling. Specimens (i.e., feces) or necropsy tissues from which the *S.*
*Dublin* was isolated may not represent the levels of resistant isolates that would be encountered during surveillance of healthy animals. It can be assumed that most of the source animals had likely received antibiotic treatment(s) prior to testing at CAHFS; however, treatment history is only rarely provided at the time of sample or carcass submission and so cannot be accurately established. Additionally, *S.*
*Dublin* is frequently isolated from multiple tissues; the specimen source isolate that was selected for further characterization was at the discretion of the diagnostician and potentially subject to lack of equal representation in the study dataset. Clinical history provided on submission forms was variable and generally limited; therefore, information regarding clinical history, management practices (including prior antimicrobial administration), and production type were not available to better analyze risk factors. This study does not provide information about the prevalence of *S*. *Dublin* among cattle in California.

## 4. Materials and Methods

### 4.1. Sample Source

*S.**Dublin* isolates were recovered from cattle specimens submitted to the California Animal Health and Food Safety Laboratory (CAHFS) between January 1993 and December 2019. Isolates were obtained from clinically ill animals either at the time of necropsy or from diarrhea samples. Approximately ten isolates per year were selected to increase the representation of the diversity of isolates over time. Criteria for inclusion included having a confirmed serovar as being *Dublin* and having an association with the clinical disease based on laboratory submission form information; furthermore, to reduce the risk of oversampling, a same location, only one isolate what collected from the same farm in the same year. Additionally, whenever possible, isolates within each year period were selected from distinct counties to avoid over-representation from particular regions in the state.

### 4.2. Salmonella Identification and Serotyping

Lyophilized or cryopreserved isolates stored at −80 °C were utilized in this study. Following a standard approach for samples submitted to CAHFS, all isolates had previously been isolated and identity confirmed by conventional aerobic culture and classical serotyping methods. Following the recovery of preserved isolates, *Salmonella* sp. identification was confirmed by matrix-assisted laser desorption–ionization mass spectrometry (MALDI-TOF; Bruker Daltonics, Fremont, CA) and serovar was confirmed using the Luminex nucleic acid bead-based suspension array and xMAP^®^ *Salmonella* serotyping assay (Luminex; Austin, TX, USA).

### 4.3. Antimicrobial Susceptibility Testing

In vitro MIC determination was performed in accordance with criteria provided in Clinical and Laboratory Standards Institute (CLSI) documents [18,37] using the Bovine/Porcine (BOPO6F) and the NARMS Gram-negative panels (Thermo Fisher Scientific, Sensititre). Antimicrobial resistance interpretation was defined using the Veterinary CLSI-defined (when available for BOPO6F panel drugs) or NARMS consensus breakpoints [17,18]. Drugs tested on the BOP6F panel include: ceftiofur (0.25–8 µg/mL), tiamulin (0.5–32 µg/mL), chlortetracycline (0.5–8 µg/mL), gentamicin (1–16 µg/mL), florfenicol (0.25–8 µg/mL), oxytetracycline (0.5–8 µg/mL), penicillin (0.12–8 µg/mL), ampicillin (0.25–16 µg/mL), danofloxacin (0.12–1 µg/mL), sulphadimethoxine (256 µg/mL), neomycin (4–32 µg/mL), trimethoprim/sulfamethoxazole (2/38 µg/mL), spectinomycin (8–64 µg/mL), tylosin (0.5–4 µg/mL), tulathromycin (1–64 µg/mL), tilmicosin (4–64 µg/mL), clindamycin (0.25–16 µg/mL), and enrofloxacin (0.12–2 µg/mL). Drugs tested on the NARMS panel include: cefoxitin (0.5–32 µg/mL), azithromycin (0.125–16 µg/mL), chloramphenicol (2–32 µg/mL), tetracycline (4–32 µg/mL), ceftriaxone (0.25–64 µg/mL), amoxicillin/clavulanic acid (1/0.5–32/16 µg/mL), ciprofloxacin (0.015–4 µg/mL), gentamicin (0.25–16 µg/mL), nalidixic acid (0.5–32 µg/mL), ceftiofur (0.12–8 µg/mL), sulfisoxazole (16–256 µg/mL), trimethoprim/sulfamethoxazole (0.12/2.38–4/76 µg/mL), ampicillin (1–32 µg/mL), and streptomycin (2–64 µg/mL). Quality control testing was performed on *Escherichia coli* ATTC 25922, *Pseudomonas aeruginosa* ATTC 27853, *Enterococcus faecalis* ATTC 29212, and *Staphylococcus aureus* ATTC 29213. 

### 4.4. Statistical Analysis

For all statistical analyses conducted, clindamycin, penicillin, sulphadimethoxine, tiamulin, tilmicosin, trimethoprim sulfamethoxazole, and tylosin in the BOPO6F panel were not included because their MIC distribution had more than 95% of isolates within one same dilution distribution or the MIC distributions were within less than 3 dilutions. Furthermore, for the BOPO6F panel, only 2 drugs could be classified using the SIR systems either because the MIC breakpoints were not contained within the range of concentrations tested or because cattle-specific clinical breakpoints for the drug and organism combination are lacking. The two drugs in the BOPO6F panel for which SIR classification was conducted were ampicillin and ceftiofur.

A table with MIC distribution for each antimicrobial was created for each drug in BOPO6F and NARMS panel. For the BOPO6F panel, sulfadimethoxine, trimethoprim sulfamethoxazole, and tylosin were not included because all of their isolates were within one MIC distribution, 246, 2, and 32 µg/mL, respectively.

Epidemiological cut-off points (ECOFFs) were calculated with three endpoint criteria (95%, 97.5%, and 99%) using ECOFFINDER version 2.1, which is based on the methodology described by Turnidge et al. (2006) [19]. Using ECOFFS, isolates with a MIC above ECOFF were categorized as resistant and those with and MIC below the ECOFF were categorized as susceptible. Assumptions and standard for using ECOFFINDER were not fulfilled by every antimicrobial tested and because of that only select ECOFFS were determined for select antimicrobial drugs [38]. Some of the standards not fulfilled, resulting in the exclusion of antimicrobial forms using ECOFFS, include that the necessity of a single peak in the putative wild-type MIC distribution and a wild-type MIC distribution that follow a log-normal distribution 3 to 5 two-fold dilutions wide.

Multiple Pearson chi-squared tests were first used to evaluate the association between the descriptive variables (year group, region, clinical signs, source group, and age group) and the binomial variable for each *S.*
*Dublin* isolate based on their classification as resistant to the referred antimicrobial drug based on CLSI breakpoints for each antimicrobial drug in the NARMS panel. No significant association between categorical variables and the azithromycin, sulfisoxazole, and trimethoprim sulfamethoxazole was observed, and they were excluded from further analysis. Following this, for each antimicrobial drug a logistic regression model in SAS was used with the antimicrobial drug resistance classification as the dependent variable and descriptive variables with a *p* value less than 0.1 from the Chi square analysis being offered to the model as explanatory variables. Variables with a *p* value greater than 0.05 were removed from the model. The Wald statistics and the Akaike information criterion (AIC) were also used for model selection and to assure a more parsimonious model was selected. The goodness of fit of each model was evaluated by performing the Hosmer–Lemeshow test. The odds ratio was calculated in the model to evaluate the odds for isolating a *S.*
*Dublin* isolate resistant to the referred drug according to different response levels of the descriptive variables. Pairwise comparisons for the least square means of the proportion of resistance isolated to relevant variables were conducted, adjusting for multiple comparisons using the Tukey–Kramer approach.

## 5. Conclusions

The high levels of AMR reported herein and elsewhere for *S.*
*Dublin* indicate that empirical antimicrobial therapy in affected cattle is not likely to be effective. Alternative intervention strategies are needed to control this pathogen in bovine herds, particularly as antimicrobial drug stewardship principles are followed, and to minimize risks of exposure in humans. Efforts to mitigate infections should focus on management practices that will reduce prevalence, prevent spread between animals, reduce the severity of infections, and support timely and effective treatments. Herd-level management strategies may include the following: the identification and separation of carrier animals to remove ongoing sources for infection and disease in susceptible animals; the rapid and accurate diagnosis of *S.*
*Dublin* in sick animals (including differentiation from BRD in septicemic animals with respiratory signs); the use of effective vaccine(s); the environmental monitoring and removal of contaminating *S.*
*Dublin*; and the judicious use of antimicrobials (guided by individual animal antimicrobial susceptibility data) to treat sick animals while preserving the efficacy of antimicrobials.

Because *S.*
*Dublin* is such a significant cattle pathogen and is not readily managed by antimicrobial therapy, future studies that can determine the prevalence of *S. Dublin* in cattle in California, both the prevalence of animals harboring the bacteria within a given operation as well as the prevalence of farms with *S. Dublin*-infected cattle will be beneficial to direct management decisions and reduce the ineffective use of antimicrobials. A better understanding of the differences in resistance profiles between isolates associated with clinical disease and antimicrobial treatment(s) and those that are recovered from healthy/asymptomatic animals would also yield valuable information.

## Figures and Tables

**Figure 1 antibiotics-11-01110-f001:**
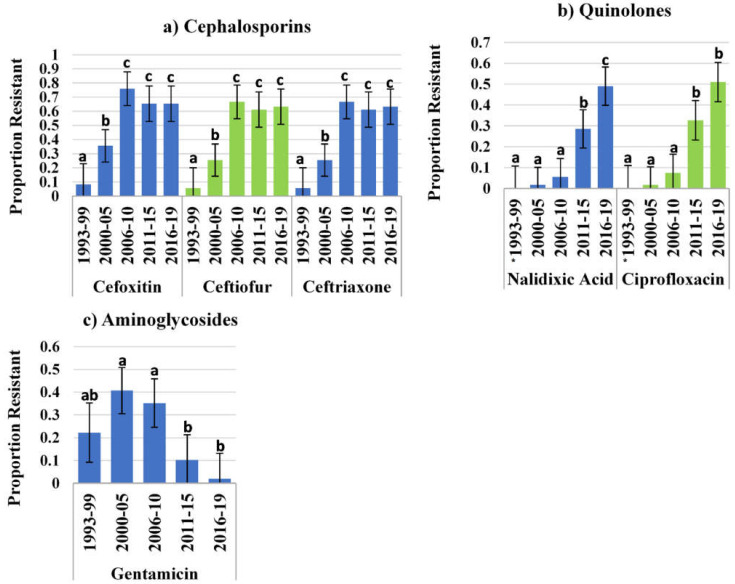
Least square means for antimicrobial resistant proportion for (**a**) cephalosporins, (**b**) quinolones and (**c**) aminoglycoside drugs. Letters are the output from Tukey-Kramer pairwise analysis, and different letters indicate a significant difference between year intervals for a referred drug. * Year interval not included in the pairwise analysis because no isolate was resistant to the referred drug during that year interval.

**Table 1 antibiotics-11-01110-t001:** Distribution of isolates by response variables year group, age group, specimen source, clinical signs, region and season.

Year Group	n	Age Group	n	Specimen Source	n	Clinical Signs	n	Region	n	Season	n
1993–1999	36	Early PW	97	Feces	98	Diarrhea	62	North	82	Winter	58
2000–2005	59	Late PW	49	Liver	66	Systemic	162	Central	132	Spring	59
2006–2010	54	Early HF	46	Lungs	64	Unknown	23	South	33	Summer	67
2011–2015	49	Adult Cow	22	Lymph node	10	-	-			Fall	62
2016–2019	49	NA *	33	Other	9	-	-			NA *	1

* Relevant detail not provided with submission.

**Table 2 antibiotics-11-01110-t002:** Distribution of minimum inhibitory concentration (MIC) and resistant for *S.*
*Dublin* isolates (n = 247) by individual drug for the BOPO6F panel. Highlighted areas in blue correspond to susceptible/intermediate classification, and the red highlighted area corresponds to resistant classification. Epidemiological cut-off points (ECOFFs) for antimicrobial drugs that fulfilled criteria for this approach are marked with a yellow highlighted asterisk * (95% endpoint criteria) and green highlighted asterisk * (99% endpoint criteria).

BOPO6F	Breakpoints (µg/mL)						% Distribution of MICs (µg/mL) ^1^									
Antimicrobial	S	I	R	%R ^2^	0.12	0.25	0.5	1	2	4	8	16	32	64	128	GAD ^5^
Ampicillin ^3^	≤0.25	0.5	≥1.0	97.1	-	-	3	19	8	3	-	-	-	-	-	67
Ceftiofur ^4^	≤2	4	≥8	46.1	-	-	26	22	4	-	44	-	-	-	-	2
Chlortetracycline	-	-	-	-	-	-	1	10	11	2	1	-	-	-	-	75
Clindamycin	-	-	-	-	-	-	-	-	-	-	1	0	-	-	-	99
Danofloxacin	-	-	-	-	81	-	14	0.5	-	-	-	-	-	-	-	0.5
Enrofloxacin	-	-	-	-	81	-	15	1	-	-	-	-	-	-	-	0
Florfenicol	-	-	-	-	-	-	-	1	-	16	4	-	-	-	-	64
Gentamicin	-	-	-	-	-	-	-	72	-	1	-	5	-	-	-	18
Neomycin	-	-	-	-	-	-	-	-	-	48	1	1	4	-	-	46
Oxytetracycline	-	-	-	-	-	-	1	-	-	2	0	-	-	-	-	77
Penicillin	-	-	-	-	-	-	-	-	-	4	15	-	-	-	-	81
Spectinomycin	-	-	-	-	-	-	-	-	-	-	-	1	77	10	-	12
Tiamulin	-	-	-	-	-	-	-	-	-	-	-	-	1	-	-	98
Tilmicosin	-	-	-	-	-	-	-	-	-	1	-	1	1	40	-	57
Tulathromycin	-	-	-	-	-	-	-	-	-	17	-	43	12 *	1 *	-	0

^1^ Distribution of the percentage of isolates with minimum inhibitory concentration (MIC). ^2^ Percent of isolates classified as resistant to the referred antimicrobial drug. ^3^ Clinical breakpoint extrapolated from bovine *Escherichia coli*, metritis. ^4^ Clinical breakpoint extrapolated from bovine *Escherichia coli*, mastitis. ^5^ Growth in all dilutions tested for the referred drug, indicating an MIC greater than the highest concentration tested.

**Table 3 antibiotics-11-01110-t003:** Distribution of minimum inhibitory concentration (MIC) and resistant for *S.*
*Dublin* isolates (n = 247) by individual drug for the NARMS panel. Highlighted areas in blue correspond to susceptible/intermediate classification, and the red highlighted area corresponds to the resistant classification. Epidemiological cut-off points (ECOFFs) for antimicrobial drugs that fulfilled criteria for this approach are marked with a yellow highlighted asterisk * (95% endpoint criteria) and a green highlighted asterisk * (99% endpoint criteria).

NARMS	Breakpoints (µg/mL)								% Distribution of MICs (µg/mL) ^1^											
Antimicrobial	S	I	R	%R ^2^	% NWT ^6^	0.015	0.03	0.06	0.12	0.25	0.5	1	2	4	8	16	32	64	256	GAD ^5^
Streptomycin	≤16	N/A	≥32	81	-	-	-	-	-	-	-	-	-	-	3	16	16	2	-	63
Tetracycline	≤4	8	≥16	76	-	-	-	-	-	-	-	-	-	24	-	1	1	-	-	74
Chloramphenicol	≤8	16	≥32	70	-	-	-	-	-	-	-	-	2	23	2	3	1	-	-	69
Ampicillin	≤8	16	≥32	67	-	-	-	-	-	-	-	22	8	2	0.5	-	0.5	-	-	67
AMC ^3^	≤8/4	16/8	≥32/16	46	-	-	-	-	-	-	-	22	9	3	16	4	1	-	-	45
Ceftiofur	≤2	4	≥8	46	-	-	-	-	-	1	22	29	2	-	3	-	-	-	-	43
Ceftriaxone	≤1	2	≥4	46	31	-	-	-	-	54	-	-	-	1	4 *	12 *	17	10	-	2
Cefoxitin	≤8	16	≥32	45	-	-	-	-	-	-	-	1	16	18	13	7	38		-	7
Gentamicin	≤4	8	≥16	23	24	-	-	-	-	11	27	4 **	1	-	-	4	-	-	-	19
Nalidixic acid	≤16	N/A	≥32	17	19	-	1	-	-	-	1	1	8	67	4 **	2	-	-	-	17
TMS ^4^	≤2/38	N/A	≥4/76	3.2	4	-	-	-	42	44	10	1	-	3	-	-	-	-	-	0
Cipropfloxacin	≤0.06	0.12–0.5	≥1	1.2	20	29	48 *	4 *	3	7	7.5	1	0.5	-	-	-	-	-	-	0
Azithromycin	≤16	N/A	≥32	0	0	-	-	-	-	-	1	1	35.5	61 *	1.5 *		-	-	-	0
Sulfisoxazole	≤256	N/A	≥512	67	-	-	-	-	-	-	-	-	-	-	-	22	8	2	1	67

^1^ Distribution of the percentage of isolates with minimum inhibitory concentration (MIC). ^2^ Percent of isolates classified as resistant to the referred antimicrobial drug. ^3^ Amoxicillin clavulanic acid; only the nominator for the fractions for this drug combination are referred to in the MIC table. ^4^ Trimethoprim sulfamethoxazole; only the nominator for the fractions for this drug combination are referred to in the MIC table. ^5^ Percent of isolates with growth in all dilutions tested for the referred drug, indicating an MIC greater than the highest concentration tested. ^6^ Percentage of isolates classified as non-WT using ECOFFs with 99% endpoint criteria. N/A indicates no intermediate breakpoint available for referenced drug.

**Table 4 antibiotics-11-01110-t004:** Epidemiological cut-off points (ECOFFs) were calculated with three endpoint criteria for antimicrobial distributions meeting the criteria for the ECOFFs method.

**Antimicrobial Drug**	**ECOFFs Endpoint Criteria%**		
	95	97.5	99
Tulathromycin_BOPO6F	32	64	64
Azithromycin_NARMS	4	4	8
Ceftriaxone_NARMS	8	16	16
Ciprofloxacin_NARMS	0.03	0.06	0.06
Gentamycin_NARMS	2	2	2
Nalidixic acid_NARMS	8	8	8
Trimethoprim-Sulfadimethoxine_NARMS	0.5	0.5	0.5

**Table 5 antibiotics-11-01110-t005:** Odds ratio for the effect of year group, region and animal age group for a *S.*
*Dublin* classified as resistant for cephalosporin drugs in the NARMS panel. *p* values in bold indicates significant odds ratio associations.

	Ceftiofur			Ceftriaxone			Cefoxitin		
Variable ^1^	OR ^2^	95% CI ^3^	*p* Value ^4^	OR ^2^	95% CI ^3^	*p* Value ^4^	OR ^2^	95% CI ^3^	*p* Value ^4^
Year Group ^5^			<0.0001			0.0004			<0.0001
1993–1999 vs. 2000–2005	0.1	(0–0.6)	0.01	0.1	(0–0.6)	0.01	0.1	(0–0.6)	0.01
1993–1999 vs. 2006–2010	0.02	(0–0.1)	<0.0001	0.0	(0–0.1)	<0.0001	0.02	(0–0.1)	<0.0001
1993–1999 vs. 2011–2015	0.02	(0–0.1)	<0.0001	0.0	(0–0.1)	<0.0001	0.02	(0–0.1)	<0.0001
1993–1999 vs. 2016–2019	0.02	(0–0.1)	<0.0001	0.0	(0–0.1)	<0.0001	0.03	(0–0.1)	<0.0001
2000–2005 vs. 2006–2010	0.1	(0–0.3)	0.0001	0.1	(0.1–0.4)	0.0001	0.1	(0–0.4)	0.0001
2000–2005 vs. 2011–2015	0.2	(0.1–0.5)	0.0007	0.2	(0.1–0.5)	0.0007	0.1	(0–0.4)	0.0001
2000–2005 vs. 2016–2019	0.2	(0.1–0.4)	0.0002	0.1	(0.1–0.4)	0.0002	0.2	(0.1–0.5)	0.001
2006–2010 vs. 2011–2015	1.3	(0.5–3.5)	0.55	1.3	(0.5–3.5)	0.55	1.0	(0.4–2.8)	0.93
2006–2010 vs. 2016–2019	1.1	(0.4–3.0)	0.78	1.1	(0.4–2.9)	0.78	1.4	(0.6–3.7)	0.41
2011–2015 vs. 2016–2019	0.8	(0.3–2.1)	0.72	0.8	(0.3–2.1)	0.72	1.4	(0.6–3.5)	0.45
**Region** ^6^			0.0004			<0.0001			0.0002
Central vs. North	3.6	(1.7–7.4)	0.0005	3.6	(1.8–7.5)	0.0005	3.7	(1.8–7.6)	0.0004
Central vs. South	0.5	(0.2–1.7)	0.31	0.6	(0.2–1.7)	0.32	0.4	(0.1–1.3)	0.15
North vs. South	0.1	(0–0.5)	0.002	0.1	(0–0.5)	0.002	0.1	(0–0.4)	0.0006

^1^ Pairwise odds ratio comparison for all response variable levels. The first response level is the reference (e.g., if OR is equal to 0.5 for year group 1993–1999 vs. 2000–2005, this means there is a 0.5 odds ratio for isolation of a *S.*
*Dublin* resistant to the referred antimicrobial in the 1993–1999 year interval when compared to the 2000–2005 year interval). ^2^ Odds ratio. ^3^ This represents the 95% confidence interval of the odds ratio. ^4^ *p* value for the odds ratio. ^5^ Year interval when *S.*
*Dublin* was isolated. ^6^ California region where *S.*
*Dublin* was isolated.

**Table 6 antibiotics-11-01110-t006:** Odds ratio for the effect of year group and region for *S.*
*Dublin* classified as resistant for tetracyclines and aminoglycosides in the NARMS panel. *p* values in bold indicate significant odds ratio associations.

Variable ^1^	Tetracycline			Gentamicin			Streptomycin		
	OR ^2^	95% CI ^3^	*p* Value ^4^	OR^2^	95% CI ^3^	*p* Value ^4^	OR ^2^	95% CI ^3^	*p* Value ^4^
**Year Group** ^5^			0.13 *			0.0004			0.42 *
**1993–1999 vs. 2000–2005**	0.4	(0.1–1.2)	0.09	0.2	(0.1–0.8)	0.01	0.7	(0.2–2.5)	0.60
**1993–1999 vs. 2006–2010**	0.2	(0.1–0.9)	0.03	0.6	(0.2–1.8)	0.38	0.6	(0.2–2.2)	0.45
**1993–1999 vs. 2011–2015**	0.2	(0–0.7)	0.01	1.8	(0.5–6.6)	0.34	0.3	(0.1–1.2)	0.08
**1993–1999 vs. 2016–2019**	0.3	(0.1–1.1)	0.07	9.6	(1.1–83.8)	0.04	0.9	(0.2–3.0)	0.83
**2000–2005 vs. 2006–2010**	0.6	(0.2–2.2)	0.50	2.3	(0.9–5.8)	0.09	0.8	(0.3–2.6)	0.79
**2000–2005 vs. 2011–2015**	0.5	(0.1–1.7)	0.26	6.8	(2.2–21.4)	0.0009	0.4	(0.1–1.4)	0.16
**2000–2005 vs. 2016–2019**	0.8	(0.2–2.7)	0.79	35.4	(4.4–282.6)	0.0008	1.2	(0.4–3.6)	0.71
**2006–2010 vs. 2011–2015**	0.7	(0.2–2.8)	0.65	3.0	(0.9–10.0)	0.07	0.4	(0.1–1.7)	0.23
**2006–2010 vs. 2016–2019**	1.3	(0.4–4.1)	0.66	15.7	(1.9–130.0)	0.01	1.4	(0.4–4.2)	0.52
**2011–2015 vs. 2016–2019**	1.8	(0.5–6.4)	0.38	5.1	(0.6–47.1)	0.14	3.2	(0.8–12.0)	0.07
**Region** ^6^			<0.0001			0.003			<0.0001
**Central vs. North**	11.4	(4.5–28.5)	<0.0001	1.2	(0.5–3.0)	0.60	3.6	(3.6–21.0)	<0.0001
**Central vs. South**	0.9	(02–3.7)	0.86	0.2	(0.1–0.6)	0.002	2.0	(0.5–7.5)	0.26
**North vs. South**	0.08	(0–0.3)	0.0007	0.2	(0.1–0.5)	0.002	0.2	(0.1–0.8)	0.02
**Age Group** ^7^			0.04						
**Cow vs. Early PW**	0.3	(0.1–0.9)	0.03	-	-	-	-	-	-
**Cow vs. Early HF**	0.1	(0–0.6)	0.01	-	-	-	-	-	-
**Cow vs. Late PW**	0.5	(0.1–1.6)	0.22	-	-	-	-	-	-
**Early PW vs. Early HF**	0.5	(0.1–1.8)	0.27	-	-	-	-	-	-
**Early PW vs. Late PW**	1.6	(0.6–4.1)	0.28	-	-	-	-	-	-
**Early HF vs. Late PW**	3.5	(0.8–15.0)	0.08	-	-	-	-	-	-
**Season** ^8^			0.04						
**Fall vs. Spring**	3.3	(1.0–11.1)	0.05	-	-	-	-	-	-
**Fall vs. Summer**	3.4	(1.3–14.4)	0.01	-	-	-	-	-	-
**Fall vs. Winter**	1.2	(0.3–4.1)	0.76	-	-	-	-	-	-
**Spring vs. Summer**	1.3	(0.5–3.7)	0.60	-	-	-	-	-	-
**Spring vs. Winter**	0.4	(0.1–1.2)	0.09	-	-	-	-	-	-
**Summer vs. Winter**	0.3	(0.08–0.9)	0.03	-	-	-	-	-	-

* Year group variable not significant for the Wald chi-square tests but was retained in the final model. ^1^ Pairwise odds ratio comparison for all response variable levels. The first response level is the reference (e.g., if OR is equal to 0.5 for year group 1993–1999 vs. 2000–2005, this means there is a 0.5 odds ratio for isolation of a *S*. *Dublin* resistant to the referred antimicrobial in the 1993–1999 year interval when compared to the 2000–2005 year interval). ^2^ Odds ratio. ^3^ This represents the 95% confidence interval of the odds ratio. ^4^ *p* value for the odds ratio or for the Wald chi-square tests for the variable in the model. ^5^ Year interval when *S.*
*Dublin* was isolated. ^6^ California region where *S.*
*Dublin* was isolated. ^7^ Age groups of animals from which *Salmonella* was isolated. Early PW (Early preweaned): ≤4 weeks of age; Late PW (Late preweaned): >4 weeks of age and ≤9 weeks of age; Early HF (Early Heifer): >9 weeks of age and ≤12 months of age; Cow: >17 months of age. ^8^ Season of the year when *S. Dublin* was isolated.

**Table 7 antibiotics-11-01110-t007:** Odds ratio for the effect of year group and region for a *S.*
*Dublin* being classified as resistant for quinolones and aminopenicillins by antimicrobial class for antimicrobial drugs in the NARMS panel. *p* values in bold indicates significant odds ratio associations.

Variable ^1^	Nalidixic Acid			Ampicillin			AMC *		
	OR ^2^	95% CI ^3^	*p* Value ^4^	OR ^2^	95% CI ^3^	*p* Value ^4^	OR ^2^	95% CI ^3^	*p* Value ^4^
Year Group ^5^			<0.0001			0.33 ****			<0.0001
**1993–1999 vs. 2000–2005**	-	-	-	0.5	(0.2–1.5)	0.25	0.1	(0–0.6)	0.01
**1993–1999 vs. 2006–2010**	-	-	-	0.3	(0.1–0.9)	0.03	0.02	(0–0.1)	<0.0001
**1993–1999 vs. 2011–2015**	-	-	-	0.7	(0.2–1.9)	0.48	0.02	(0–0.1)	<0.0001
**1993–1999 vs. 2016–2019**	-	-	-	0.6	(0.2–1.6)	0.30	0.02	(0–0.1)	<0.0001
**2000–2005 vs. 2006–2010**	0.4	(0–4.9)	0.49	0.6	(0.2–1.6)	0.27	0.1	(0.1–0.4)	0.0001
**2000–2005 vs. 2011–2015**	0.04	(0–0.3)	0.004	1.3	(0.5–3.3)	0.62	0.2	(0.1–0.5)	0.0007
**2000–2005 vs. 2016–2019**	0.01	(0–1.3)	0.0001	1.1	(0.4–2.7)	0.88	0.2	(0.1–0.4)	0.0002
**2006–2010 vs. 2011–2015**	0.1	(0–0.5)	0.008	2.2	(0.8–6.3)	0.12	1.3	(0.5–3.5)	0.55
**2006–2010 vs. 2016–2019**	0.04	(0–0.2)	0.0002	1.9	(0.7–5.2)	0.21	1.1	(0.4–2.9)	0.78
**2011–2015 vs. 2016–2019**	0.3	(0.1–0.9)	0.04	0.8	(0.3–2.2)	0.72	0.8	(0.3–2.1)	0.73
**Region ^6^**			0.001			<0.0001			0.0004
**Central vs. North**	7.5	(2.4–23.0)	0.0005	5.1	(2.6–10.2)	<0.0001	3.6	(1.7–7.5)	0.0005
**Central vs. South**	0.6	(0.1–2.7)	0.5	0.8	(0.3–2.5)	0.75	0.6	(0.2–1.7)	0.32
**North vs. South**	0.08	(0–0.5)	0.005	0.2	(0.1–0.5)	0.001	0.2	(0.05–0.5)	0.002

* Amoxicillin clavulanic acid. ** Year group variable not significant for the Wald chi-square tests but was retained in the final model. ^1^ Pairwise odds ratio comparison for all response variable levels. The first response level is the reference (e.g., if OR is equal to 0.5 for year group 1993–1999 vs. 2000–2005, this means there is a 0.5 odds ratio for isolation of a *S.*
*Dublin* resistant to the referred antimicrobial in the 1993–1999 year interval when compared to the 2000–2005 year interval). ^2^ Odds ratio. ^3^ This represents the 95% confidence interval of the odds ratio. ^4^ *p* value for the odds ratio or for the Wald chi-square tests for the variable in the model. ^5^ Year interval when *S.*
*Dublin* was isolated. ^6^ California region where *S.*
*Dublin* was isolated. 7 Season of the year when *S.*
*Dublin* was isolated.

**Table 8 antibiotics-11-01110-t008:** Odds ratio for the effect of year group and region for a *S.*
*Dublin* classified as resistant for chloramphenicol in the NARMS panel. *p* values in bold indicate significant odds ratio associations.

Variable ^1^	Chloramphenicol		
	OR ^2^	95% CI ^3^	*p* Value ^4^
Year Group ^5^			<0.0001
**1993–1999 vs. 2000–2005**	0.1	(0–0.5)	0.001
**1993–1999 vs. 2006–2010**	0.06	(0–0.2)	<0.0001
**1993–1999 vs. 2011–2015**	0.04	(0–0.2)	<0.0001
**1993–1999 vs. 2016–2019**	0.06	(0–0.2)	<0.0001
**2000–2005 vs. 2006–2010**	0.4	(0.1–1.3)	0.13
**2000–2005 vs. 2011–2015**	0.27	(0.1–1.0)	0.04
**2000–2005 vs. 2016–2019**	0.4	(0.1–1.3)	0.11
**2006–2010 vs. 2011–2015**	0.7	(0.2–2.7)	0.59
**2006–2010 vs. 2016–2019**	1.0	(0.3–3.5)	0.95
**2011–2015 vs. 2016–2019**	1.5	(0.4–5.5)	0.54
**Region ^6^**			<0.0001
**Central vs. North**	17.911	(6.6–48.5)	<0.0001
**Central vs. South**	0.660	(0.2–2.5)	0.54
**North vs. South**	0.037	(0–0.2)	<0.0001
**Season** ^7^			0.01
**Fall vs. Spring**	5.5	(1.7–17.7)	0.004
**Fall vs. Summer**	3.6	(1.1–11.1)	0.03
**Fall vs. Winter**	1.3	(0.4–4.2)	0.68
**Spring vs. Summer**	0.6	(0.2–1.7)	0.37
**Spring vs. Winter**	0.2	(0.1–0.7)	0.01
**Summer vs. Winter**	0.3	(0.1–1.1)	0.08

^1^ Pairwise odds ratio comparison for all response variable levels. The first response level is the reference (e.g., if OR is equal to 0.5 for year group 1993–1999 vs. 2000–2005, this means there is a 0.5 odds ratio for isolation of a *Salmonella*
*Dublin* resistant to the referred antimicrobial in the 1993–1999 year interval when compared to the 2000–2005 year interval). ^2^ Odds ratio. ^3^ This represents the 95% confidence interval of the odds ratio. ^4^ *p* value for the odds ratio or for the Wald chi-square tests for the variable in the model. ^5^ Year interval when *S.*
*Dublin* was isolated. ^6^ California region where *S. *Dublin** was isolated. ^7^ Season of the year when *S.*
*Dublin* was isolated.

**Table 9 antibiotics-11-01110-t009:** Odds ratio for the effect of year group, region and age group for a *S.*
*Dublin* being classified as multidrug resistant. *p* values in bold indicate significant odds ratio associations.

Variable ^1^	MDR by Class *		
	OR ^2^	95% CI ^3^	*p* Value ^4^
Year Group ^5^			0.03
**1993–1999 vs. 2000–2005**	0.3	(0.1–0.9)	0.04
**1993–1999 vs. 2006–2010**	0.1	(0–0.5)	0.006
**1993–1999 vs. 2011–2015**	0.1	(0–0.5)	0.05
**1993–1999 vs. 2016–2019**	0.2	(0–0.7)	0.02
**2000–2005 vs. 2006–2010**	0.5	(0.1–1.7)	0.28
**2000–2005 vs. 2011–2015**	0.5	(0.1–1.6)	0.24
**2000–2005 vs. 2016–2019**	0.7	(0.2–2.4)	0.64
**2006–2010 vs. 2011–2015**	0.9	(0.2–3.6)	0.92
**2006–2010 vs. 2016–2019**	1.5	(0.4–5.1)	0.49
**2011–2015 vs. 2016–2019**	1.6	(0.4–5.9)	0.46
**Region ^6^**			<0.0001
**Central vs. North**	12.4	(4.8–32.6)	<0.0001
**Central vs. South**	0.5	(0.1–2.4)	0.34
**North vs. South**	0.04	(0–0.2)	0.0002
**Age Group ^7^**			0.03
**Cow vs. Early PW**	0.3	(0.1–0.9)	0.02
**Cow vs. Early HF**	0.1	(0–0.7)	0.01
**Cow vs. Late PW**	0.6	(0.2–2.0)	0.41
**Early PW vs. Early HF**	0.5	(0.1–2.0)	0.35
**Early PW vs. Late PW**	2.1	(0.8–5.5)	0.10
**Early HF vs. Late PW**	4.1	(0.98–18)	0.052

* Binomial multidrug resistance where “1” are *Salmonella* resistant to three or more different drug classes, and “0” is all other circumstances. ^1^ Pairwise odds ratio comparison for all response variable levels. The first response level is the reference (e.g., if OR is equal to 0.5 for year group 1993–99 vs. 2000–05, this means there is a 0.5 odds ratio for isolation of a *S.*
*Dublin* resistant to the referred antimicrobial in the 1993–1999 year interval when compared to the 2000–2005 year interval). ^2^ Odds ratio. ^3^ This represents the 95% confidence interval of the odds ratio. ^4^ *p* value for the odds ratio. ^5^ Year interval when *S.*
*Dublin* was isolated. ^6^ California region where *S.*
*Dublin* was isolated. ^7^ Age groups of animals from which *S.*
*Dublin* was isolated. Early PW (Early preweaned): ≤4 weeks of age; Late PW (Late preweaned): >4 weeks of age and ≤9 weeks of age; Early HF (Early Heifer): >9 weeks of age and ≤12 months of age; Cow: >17 months of age.

## Data Availability

This study was funded by the Antimicrobial Use and Stewardship (AUS) Program of the California Department of Food and Agriculture and is subject to California Food and Agricultural Code (FAC) Sections 14400 to 14408. FAC Section 14407 requires that data collected be held confidential to prevent the individual identification of a farm or business; as such, raw data from this study is not able to be shared.

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
