# Peer review of "Salmonella enterica Serovar Dublin from Cattle in California from 1993–2019: Antimicrobial Resistance Trends of Clinical Relevance"

_antibiotics, 2022, doi:10.3390/antibiotics11081110_

Round 1

Reviewer 1 Report

The abuse and misuse of antibiotics in animals contributes to the rise of antibiotic resistance. Some types of bacteria that cause serious human infections are already resistant to most or all available treatments and there are very few promising alternatives under investigation. Due to the increased prevalence of Salmonella Dublin in animals and clinical isolates of human origin, studies such as the one presented here are important to determine the levels of resistance of this microorganism to antibiotics commonly used in livestock and also commonly used in human clinic.

Some remarks regarding the presentation of the work would be:

-      In section 2.1, under Results, a description of the selected isolates is given and the criteria for their classification are defined. In addition, it revisits issues that have already been included in section 4, such as the selection period of the isolates or the procedures used for the antimicrobial susceptibility testing. In my opinion, the section 2.1 should be included in the Material and Method section.

-      The authors include isolates with a susceptible and intermediate clinical category in the same category (Tables 2a and 2b). In my opinion, isolates considered to have intermediate or reduced susceptibility to an antibiotic should be included in the resistant category, since their use should be restricted.

-      In the section on statistical analysis, the use of the non-parametric Wilcoxon test is not mentioned.

-      The association with the season of the year has only been considered in table 4d as it is the only one in which significant results are obtained. I consider it necessary to include the results also in tables 4a, 4b and 4c, although the values obtained are not significant.

-      The authors use the abbreviation “S. Dublin” and sometimes “Salmonella Dublin”. They should unify the criteria. I recommend revising the entire text.

-      Line 79: Salmonella instead of Salmonella

-      Line 145: Wilcoxon instead Wilcoxan

-      Line 160: Distribution instead di stribution

-      Table 4a, second row: 1993-1999 instead 1993-1919

-      In the footnotes of tables 4a to 4c and table 5 it should be eliminated “Early PW (Early preweaned): ≤ then 4 weeks of age; Late PW (Late preweaned): > 4 weeks of age and ≤ 9 weeks of age; Early HF (Early heifer): > 9 weeks of age and ≤ 12 months of age; Cow: > 17 months of age.” since this classification by age groups has been described in Materials and Methods.

-      Line 196: Table 4b does not include information on phenicols. This information is in table 4d. The title of that table should be corrected.

-      Lines 245 and 246: “There was a significantly lower OR for MDR in the early years 1993-99 compared to all later year intervals (Table 5). The MDR profiles are covered in the accompanying third manuscript.” I consider it necessary to include in the manuscript a small summary or table including at least the number of MDR isolates, i.e. with resistance to 3 or more different groups of antibiotics.

-      Line 390: Representation instead representaiton

-      Line 438: Salmonella Dublin or S. Dublin instead Salmonella Dublin

-      The format of bibliographic citations should conform to the journal's recommendations.

Reviewer 2 Report

The manuscript entitled Salmonella enterica serovar Dublin from cattle in California from 1993-2019: Part I. Antimicrobial resistance trends of clinical relevance submitted by Fritz et al. investigates the minimum inhibitory concentrations and antimicrobial susceptibility trends to antimicrobials of clinical relevance to both veterinary and human medicine in S. Dublin isolates recovered from cattle in California over a 27-year timeframe. The manuscript is generally within the scope of Antibiotics, however would benefit from editing and clarification of specific pointsHowever, some important remarks should be addressed before publication. First, the results are not properly presented, and the Authors should pay more attention to the quality of data presentation. The tables and figures must be improved before publication. Other review comments:

Line 46: double space

Line 49: The Authors state that antibiotic treatment is controversial, however, there is a high risk of mortality without timely administration of effective antibiotics. In my opinion, the Authors should briefly state which antibiotics are used in the treatment of Salomnella spp. infections. 

Lina 74: The purpose and novelty of this study must be better explained. 

Line 86: The groups of participants with n=0 shouldn’t be included in the table (e.g. Late HF). The authors should explain each abbreviation used in the table. All figures and tables in the manuscript should be self-explanatory. 

Line 113, 126: The quality of Table 2a and 2b is too low. The tables look like some random tables copied from the excel document. There are too many different fonts, too many blank cells in the tables. 

Table 4a-4d: These tables are difficult to read. All p-values in each table should have identical number of decimal digits. I suggest selecting the most important results and include just these data in the main manuscript. The rest of the data should be transferred into the supplementary material. 
